# Design of Power Amplifiers for BDS-3 Terminal Based on InGaP/GaAs HBT MMIC and LGA Technology

**DOI:** 10.3390/mi14111995

**Published:** 2023-10-27

**Authors:** Zhenbing Li, Junjie Huang, Jinrong Zhang, Shilin Jia, Haoyang Sun, Gang Li, Guangjun Wen

**Affiliations:** 1School of Information and Communication Engineering, University of Electronic Science and Technology of China, Chengdu 611731, China; thomaslizhenbing@163.com (Z.L.); hjj_rfic@163.com (J.H.); zhangjinrong1220@163.com (J.Z.); 13400293776@163.com (S.J.); sunhaoyang21010107@163.com (H.S.); 2School of Information and Software Engineering, University of Electronic Science and Technology of China, Chengdu 610054, China; ligangpm@uestc.edu.cn

**Keywords:** analog pre-distortion, BDS-3 terminal, class-F power amplifier, compatible power amplifier chips, short message function, temperature-insensitive adaptive bias

## Abstract

With the development and popularization of the Beidou-3 navigation satellite system (BDS-3), to ensure its unique short message function, it is necessary to integrate a radio frequency (RF) transmitting circuit with high performance in the BDS-3 terminal. As the key device in an RF transmitting circuit, the RF power amplifier (PA) largely determines the comprehensive performance of the circuit with its transmission power, efficiency, linearity, and integration. Therefore, in this paper, an L-band highly integrated PA chip compatible with 3 W and 5 W output power is designed in InGaP/GaAs heterojunction bipolar transistor (HBT) technology combined with temperature-insensitive adaptive bias technology, class-F harmonic suppression technology, analog pre-distortion technology, temperature-insensitive adaptive power detection technology, and land grid array (LGA) packaging technology. Additionally, three auxiliary platforms are proposed, dedicated to the simulation and optimization of the same type of PA designs. The simulation results show that at the supply voltage of 5 V and 3.5 V, the linear gain of the PA chip reaches 39.4 dB and 38.7 dB, respectively; the output power at 1 dB compression point (P1dB) reaches 37.5 dBm and 35.1 dBm, respectively; the saturated output power (Psat) reaches 38.2 dBm and 36.2 dBm, respectively; the power added efficiency (PAE) reaches 51.7% and 48.2%, respectively; and the higher harmonic suppression ratios are less than −62 dBc and −65 dBc, respectively. The size of the PA chip is only 6 × 4 × 1 mm^3^. The results also show that the PA chip has high gain, high efficiency, and high linearity under both output power conditions, which has obvious advantages over similar PA chip designs and can meet the short message function of the BDS-3 terminal in various application scenarios.

## 1. Introduction

At present, there are four major navigation satellite systems worldwide [1,2,3,4], namely, the Global Positioning System (GPS) developed by the United States, with the widest coverage area in the world [1]; the Global Navigation Satellite System (GLONASS) developed by Russia, with anti-interference ability [2]; the Global Navigation Satellite System (Galileo) developed in Europe, with excellent positioning accuracy [3]; and the BeiDou Navigation Satellite System (BDS) developed in China [4]. Compared with the other three navigation satellite systems, BDS can achieve reliable security and high-precision positioning with its unique short message communication function [5]. In some environments with no or low signal coverage, multiple BDS terminals can still communicate with each other via satellite, playing a pivotal role in satellite phone calls, earthquake rescues, outdoor locating, and other fields [6]. Notably, in a BDS terminal, the PA chip is the core device in realizing the short message function, and its performance directly affects the communication quality of such function.

Restricted by factors such as satellite reception sensitivity, limited battery capacity of wireless terminals, and the trend of terminal miniaturization, PA chips of satellite communication terminals are required to have high transmission power, high energy conversion efficiency, and high integration level [7,8,9,10,11,12,13,14,15,16]. In terms of transmission power, many scholars have made different design attempts, and works [7,8,9] have achieved relatively outstanding output power within their respective designed frequency bands. For instance, in [7], based on the GaN-Si process, a PA chip is designed in a three-stage balanced PA structure, achieving an output of 41 dBm in the K-band. In [8], the design achieves a 45.2 dBm and 48.6 dBm output in Ku-band, which is at the leading level in the present industry. However, modern satellite communication systems not only require a high output power of the terminal PA chip but also its efficiency to reduce power consumption and increase the service life of battery-powered wireless terminals. Some works have ensured the output power of their respective PA chips while also taking into account the efficiency [10,11,12]. For example, in [10], an elliptical filter structure is used based on the GaN process to design the output matching network of the PA chip, achieving a Psat of 41.1–44.1 dBm and a PAE of 19–40% in the 6–18 GHz frequency band. Authors in [11] developed a PA chip with high power and efficiency, which has an average output power of 42 dBm and a PAE of 41% in 17–20.5 GHz. Nevertheless, there is also a growing demand for the miniaturization of PA chips in modern satellite communication terminals. Therefore, it is necessary to design PA chips with not only high output power and efficiency but also high integration levels [13,14,15,16]. For example, the authors in [13] developed a highly integrated PA chip based on the GaN pHEMT (Pseudomorphic High Electron Mobility Transistor) process, and coupled lines were used to obtain the matching network. Within 10.5–15.5 GHz, a Psat of 38.6–39.9 dBm, a PAE of 35–41%, and a total chip size of only 2.7 × 2.4 mm^2^ is achieved. In another work [16], a very-small-aperture terminal (VSAT) for Ku-band satellite communication is designed. Its size reaches only 2.7 × 1.6 mm^2^, with a Psat of 35.6 dBm at 27.5–29.5 GHz and a PAE of 22.7%.

For BDS, its communication frequency band is in the L band, and due to the short message function requirements of the system, the output power of the PA chip in the battery-powered wireless terminal is required to reach at least 3 W and 5 W in some special application scenarios (such as mountainous areas, canyons, and other special areas) [17]. While the PA chip is miniaturized and fully functional, other performance parameters should be at a high level in the industry. RF3183 [18] and SKY77354 [19] are L-band PA chips launched by RFMD Co., Ltd. of Greensboro, NC, USA and Skyworks Co., Ltd. of Woburn, MA, USA, respectively. The efficiency of these two PA chips is up to 50% and 53%, respectively, which has reached a high level in the industry, but their output power makes it difficult to meet the requirements of BDS terminals. In 2013, Innotion Co., Ltd. of Suzhou, China and Creotech Co., Ltd. of Xiamen, China introduced YP163137 [20] and LXK6618 [21]. Those two PA chips both have high output power and are small in size, which can basically meet the requirements of BDS terminals, but their PAE is just 37% and 40%, respectively, which poses a challenge to the battery capacity of BDS terminals. Meanwhile, many universities have carried out corresponding research works on L-band PA chips. In [22], an L-band PA chip for BDS handheld terminals is designed. Using double-link power synthesis technology, its Psat reaches 37.8 dBm, and the PAE reaches 43%. However, this PA combines two 3 W MMIC (Monolithic Microwave Integrated Circuits) Die to output a synthesis power of 5 W, which not only leads to a large chip size (10 × 10 × 1 mm^3^) but also a gain of only 30 dB. In [23], a high-efficiency, high-linearity PA chip for BDS handheld terminals is proposed. Adopting a completely symmetrical double-link power synthesis architecture based on a class-F output matching network, its P1dB reaches 37 dBm, Psat reaches 38 dBm, PAE reaches 45%, and the small signal gain reaches 30 dB. Despite its advantages in chip size (6 × 6 × 0.82 mm^3^) and PAE compared to [22], its gain and integration still need improvement. In 2021, the authors of [24] designed a 5 W high-power, high-linearity L-band PA chip for BDS terminals with P1dB of 37.3 dBm and Psat of 37.8 dBm, but the PAE is only 36.5%, which makes it hard to meet the requirements of PA chips for BDS-3 terminals.

In view of the problems such as low output power, low efficiency, low gain, and low integration of the PA chip in the navigation satellite system terminals mentioned above, to meet the short message function of BDS-3 terminals (battery-powered wireless terminals), this paper introduces a 3 W and 5 W compatible PA chip operating at 1.6–1.65 GHz based on InGaP/GaAs HBT technology for BDS-3 terminals. The simulation results show that the functions and performance of the designed PA chip are at the highest level in the industry with high gain, high efficiency, and high linearity under two output power conditions (P1dB reaches 3 W and P1dB reaches 5 W), which can meet the short message function of BDS terminals and the requirements of different application scenarios. The main innovations and contributions of this work compared to others are:A compatible PA chip is designed based on InGaP/GaAs HBT technology. Simulation results show that by changing only the circuit supply voltage, P1dB switches from 3 W to 5 W with high efficiency and linearity, respectively. Different from traditional multi-mode PAs that require reconfigurable circuits for mode switching [25,26], the PA chip proposed in this paper is simpler and more reliable. And the quiescent current is as low as 280 mA (@5 W) and 262 mA (@3 W), which can meet the short message function of battery-powered BDS terminals in different application scenarios;It is different from traditional PA modules in the traditional base stations or terminals that require multi-cascade PA chips to achieve high gain and high output power [22,23]. The proposed PA chip can directly achieve a linear gain of 39.4 dB, P1dB of up to 37.5 dBm, and Psat of 38.2 dBm, which is currently at the highest level in the industry. The total chip size is only 6 × 4 × 1 mm^3^ with good thermal characteristics, and its integration level exceeds similar PA products;The proposed PA chip integrates a temperature-insensitive adaptive power detection circuit module and ESD (Electro-Static Discharge) electrostatic protection circuits without affecting the electromagnetic performance, thermal performance, or size, compared with a similar PA design [24]; the PA chip proposed in this paper is more complete and reliable, which provides a strong guarantee for the accurate power monitoring and controlling of terminals as well as its yield rate;A class-F output matching circuit co-simulation design platform, a 3 W and 5 W PA compatibility design platform both based on ADS (Advanced Design System) and HFSS (High-Frequency Simulator Structure), and a thermal simulation auxiliary design platform based on ANSYS are proposed, which can all be used to improve the design and optimization efficiency of similar PA products.

## 2. InGaP/GaAs HBT Technology and Design Goal

Although the first-generation semiconductor materials such as silicon (Si), germanium (Ge), and other elements have broken the limitations of bulky electron tubes, the breakdown voltage is low, resulting in the device only being used in application scenarios with low transmit power [27]. Because of higher requirements of output power, the second-generation III-V cluster compound semiconductor materials appear, such as gallium arsenide (GaAs) and indium phosphide (InP). With their excellent carrier transport capacity and high breakdown voltage, devices can meet the requirements of higher frequency, higher rate, and higher power in communication systems and are thus favored by satellite communications, aerospace communications, mobile communications, electronic consumption, smart Internet of Things systems, and other markets [28]. Research results show that when GaAs are doped with InGaP, the lattice matches well, the band gap of the second-generation semiconductor material is improved, and the current gain is increased due to the small band spacing. On the other hand, high-concentration doped emitter material passivates the base surface of HBT and reduces the composite current effect. Therefore, InGaP/GaAs HBT technology is often used for the MMICs to obtain both high output power density and low-noise characteristics with single-voltage operation in the present industry [29,30], and the HBTs are more suitable for high yield rate and low cost than pHEMTs for L-band high-power applications. With the continuous innovation of semiconductor technologies, third-generation semiconductor materials with wide bandgap have been developed [31,32,33,34], and even fourth-generation semiconductor materials with ultra-wide bandgap have become a research hotspot [35]. However, considering the cost performance in the L-band, InGaP/GaAs HBT technology is finally chosen to be used in this paper to realize the PA chip for BDS terminals. The schematic diagram of the stacked structure of InGaP/GaAs HBT used in this paper is shown in Figure 1.

This HBT technology was used because BDS terminals require high power, high efficiency, high linearity, and high integration of PA chips. Considering such requirements, this PA chip needs to operate in the L-band and output 5 W (@5 V) and 3 W (@3.5 V) with a gain greater than 37 dB, which meets the short message function requirements of BDS terminals without the need of driver amplifiers (transmit power reaches 3 W, and 5 W in some special application scenarios). Its PAE needs to be greater than 40%, which achieves high efficiency and meets the power consumption requirements of BDS terminals. The second, third, and fifth harmonic suppression ratios should be less than −60 dBc, and the AM-PM (Amplitude Modulation-Phase Modulation) distortion should be less than ±2.5°. The third-order intermodulation distortion ratio should be less than −25 dBc, achieving high linearity of BDS terminals. There should also be built-in matching circuits, adaptive bias circuits, a power detection circuit, ESD electrostatic protection circuits, and other functional modules with a chip size of less than 25 mm^3^ to achieve high integration of BDS terminals.

Figure 2 shows the volt–ampere characteristic curve of HBT, and the $V_{Knee}$ of the HBT is 0.5 V. The substrate thickness is as thin as 75 μ and helps the reduction in the thermal resistance. The HBT channel length is 3 μm, and the thicknesses of the contact metal of the base, collector, and emitter are 0.3 μm, 0.3 μm, and 0.055 μm, respectively. Typical DC characteristics are as follows—a DC current gain of 100, a collector–emitter breakdown voltage of 14 V, and a collector–base breakdown voltage of 25 V. A three-finger HBT with an emitter size of 3 × 40 × 3 μm^2^ offers a 35 GHz f_cutoff_ (cutoff frequency) and a 72 GHz f_max_osc_ (maximum oscillation frequency).

## 3. Terminal PA Chip Design

### 3.1. PA Chip Circuit Design

Because the PA in battery-powered wireless BDS terminals should have an output power of at least 3 W, and 5 W in some special application scenarios, the overall circuit architecture design purpose of the 3 W/5 W compatible PA chip is to achieve different output powers by changing the circuit supply voltage without altering any circuit and layout design while keeping the efficiency, linearity, and other indicators under unified requirements. In other words, the P1dB of the PA chip can be switched from 3 W to 5 W by changing only the circuit supply voltage. According to the design goals, based on the RF characteristics of the HBT device, the PA uses a three-stage cascaded amplification circuit structure to achieve a gain greater than 37 dB. The circuit architecture of the proposed 3 W/5 W compatible BDS terminal PA chip is shown in Figure 3, which is composed of an MMIC Die and an LGA PCB. Each HBT in any amplification stage is connected in series with a resistor and a capacitor in parallel to form a cell, which improves the RF stability of the PA. The output stage adopts 36 HBT cells in parallel to achieve P1dB greater than 35 dBm (@3.5 V) and 37 dBm (@5 V) and gain greater than 12 dB. The driver stage (intermediate stage) uses eight HBT cells in parallel to provide a linear output power greater than 24 dBm (@3.5 V) and 26 dBm (@5 V) with a gain greater than 13 dB; the input stage adopts two HBT cells in parallel to provide a linear output power greater than 13 dBm (@3.5 V) and 14 dBm (@5 V) with a gain greater than 13 dB.

The MMIC Die of the 3 W/5 W compatible BDS terminal PA chip proposed in this paper is shown in Figure 4, including three stages of amplification circuits, three stages of temperature-insensitive adaptive bias circuits, partial matching circuits between each amplification stage (except an output matching circuit), a partial temperature-insensitive adaptive power detection circuit, and ESD electrostatic protection circuits. The designed LGA PCB is shown in Figure 5a, including RF chokes, bypass capacitors, partial matching network circuits between each stage, a partial temperature-insensitive adaptive power detection circuit, and a class-F output matching network circuit with harmonic suppression function. The 3D top and bottom views of the LGA package design are shown in Figure 5b and Figure 5c, respectively.

According to the V-I characteristic curve shown in Figure 2, the bias point of the HBT device differs little when the collector–emitter supply voltage changes from 3.5 V (@3 W) to 5 V (@5 W). Furthermore, according to Equation (2), the optimal load impedances corresponding to the two output powers are close, which provides conditions for the 3 W/5 W compatibility design of the PA chip. In this paper, the 3 W/5 W compatibility design of the PA chip is realized by using the temperature-insensitive adaptive bias circuit technology, class-F harmonic suppression technology, and the PA chip compatibility optimization design platform proposed in Section 3.7. Additionally, key technologies such as analog predistortion and temperature-insensitive adaptive power detection technology are combined to improve the overall performance of the PA chip. The circuit design of each module of the PA chip will be explained separately according to the above key technologies (in order to avoid repeating, only the 5 W BDS terminal PA chip is elaborated).

### 3.2. Temperature-Insensitive Adaptive Bias Circuit Design

As the input power (RFin) and ambient temperature vary, the offset of the static operating point of the PA will cause serious problems, such as phase distortion and gain compression, which will eventually deteriorate the linearity of the PA. A temperature-insensitive adaptive bias circuit is a functional circuit module that can effectively improve PA linearity; therefore, in order to improve the linearity of the BDS terminal PA chip, a highly integrated on-chip temperature-insensitive adaptive bias circuit shown in Figure 6 is used to provide temperature-insensitive adaptive bias points for each amplification stage of the PA chip [36]. The transistor HBT0 in Figure 6 represents the amplification circuit of each stage in the PA, and the temperature-insensitive adaptive bias circuit is connected to the base electrode of HBT in each amplification stage through a ballast resistor R1. The voltage relationship between each node is as follows:(1)$$V_{be\_HBT0}=V_{B}-{R_{1}I}_{b}-V_{be\_HBT1}$$

Due to the rectification characteristics of the base–emitter diode, the base–emitter voltage of HBT0, V_be_HBT0_ varies according to input power RFin or temperature. Through this temperature-insensitive adaptive bias circuit, a portion of the RF signal is leaked to the bias circuit through resistor R1. Since the base–emitter diode of HBT1 also has a rectification function, the base–emitter voltage of HBT1, V_be_HBT1_, also varies according to input power or temperature. As can be seen in Equation (1), the change of V_be_HBT1_ compensates that of V_be_HBT0_, which stabilizes the bias point of transistor HBT0, effectively avoiding problems such as phase distortion and gain compression and improving the temperature insensitivity and linearity of the PA chip. The structure of the designed bias circuit ensures that the RF signal leaked into the bias circuit will be short-circuited to ground through the bypass capacitor C1, protecting the power supply port while stabilizing the potential at point B, ensuring the compensation stability of base–emitter voltage V_be_HBT1_ of HBT1 for V_be_HBT0_ of HBT0. Additionally, the ballast resistor R1 can effectively limit the current from the bias circuit to the base electrode of HBT0, thereby further improving the temperature insensitivity and linearity of the PA chip.

### 3.3. Design of Class-F Output Matching Network with Harmonic Suppression Characteristics

A PA often operates in a saturated state, which generates a large number of harmonic components, among which higher harmonic components such as second-, third-, and fifth-order harmonics have the greatest impact on the PAE of the PA. A class-F output matching network can effectively suppress the effect of higher harmonics on the PAE of the PA. Therefore, a class-F output matching network circuit with harmonic suppression characteristics is proposed in this paper to improve the energy conversion efficiency of the PA chip. The structure is shown in Figure 7, which realizes fundamental impedance matching while being a short circuit (low impedance) to the second harmonic component and an open circuit (high impedance) to the third and fifth harmonic components.

#### 3.3.1. Class-F Output Matching Network Design Theory

The output power, supply voltage, and intermediate impedance of the PA have the following relationship:(2)$$Z_{1}=\frac{{\left(V_{ce}-V_{Knee}\right)}^{2}}{2P_{out}}$$
where $V_{ce}$ is the collector–emitter voltage of the transistor and $V_{Knee}$ is the knee point voltage of the transistor. The intermediate impedance shall meet the following requirements:(3)$$Z_{2}=\sqrt{Z_{Load}Z_{1}}$$
where $Z_{Load}$ is 50 Ω for the BDS terminal PA chip designed in this paper.

The impedance conversion ratio is defined as
(4)$$m_{1}=\frac{Z_{Load}}{Z_{2}}$$
(5)$$m_{2}=\frac{Z_{2}}{Z_{1}}$$

When the two-stage LC matching network converts the load impedance from $Z_{Load}$ to $Z_{1}$, the capacitance and inductance of the third and fifth harmonic tuning branches need to meet the following equations:(6)$$L_{51}=\frac{Z_{2}\sqrt{m_{1}-1}}{\omega }$$
(7)$$L_{31}=\frac{Z_{1}\sqrt{m_{2}-1}}{\omega }$$
(8)$$C_{5}=\frac{24\sqrt{m_{1}-1}}{25Z_{Load}\omega }$$
(9)$$C_{3}=\frac{8\sqrt{m_{2}-1}}{9Z_{2}\omega }$$
(10)$$L_{52}=\frac{Z_{Load}}{24\omega \sqrt{m_{1}-1}}$$
(11)$$L_{32}=\frac{Z_{2}}{8\omega \sqrt{m_{1}-1}}$$

Meanwhile, in the second harmonic tuning branch, the series resonant network $L_{2}$ can be calculated based on the value of $C_{2}$:(12)$$L_{2}=\frac{1}{4\omega^{2}C_{2}}$$

According to the component values of the class-F matching circuit calculated by Equations (2)–(12), the results of the fundamental impedance, the second, third, and fifth harmonic impedances are distributed on the Smith chart shown in Figure 8. In the figure, it can be seen that the output matching network is well matched for the fundamental wave, presenting an absolute short circuit for the second harmonic and relatively high impedance for the third and fifth harmonics. According to the simulation results, a class-F output matching network with harmonic suppression characteristics designed based on the actual layout and SPICE model of Murata components improves the PAE of the PA chip by about 2%.

#### 3.3.2. Class-F Output Matching Network High-Efficiency Design Platform

Due to the problems of a class-F output matching network with harmonic suppression characteristics, such as complex calculations, large differences between ideal and actual components and circuit layout, and easy deterioration of the output standing wave ratio of the PA chip when the imaginary part of the output impedance is non-negligible, an efficient design platform and simulation optimization method is proposed in this paper based on the joint implementation of ADS + HFSS. To elaborate, a platform is proposed for designing class-F output matching circuits based on ADS + HFSS, whose architecture is shown in Figure 9. 

Firstly, a small signal simulation design platform based on Equations (2)–(12) is built on ADS. Through this platform, the optimal output impedance (including real and imaginary parts) can be obtained based on load pull, and the ideal parameters of each component in the class-F output matching circuit can be calculated. Then, according to the calculated ideal values and the PA size requirements, an output circuit model is built in HFSS with the ideal capacitance values replaced by Murata capacitances and ideal inductance values replaced by microstrip lines, where specific parts are set variables that can be optimized. Therefore, in order to simulate and optimize the variables set above, the S parameters of the ideal output matching circuit are first obtained in ADS and exported as an SnP file into HFSS, and then the optimization goal can be set as the difference between the S parameters of HFSS output circuit model and that of ADS output circuit model (imported SnP file) equals 0. After the model optimization is completed, the S parameter results in HFSS are exported as an SnP file into the LSSP (Large Signal S Parameter) and Harmonic Balance simulation platform in ADS to verify the overall performance of the PA chip. This design method can be successfully used in class-F output matching circuit designs, except when the microstrip lines occasionally do not meet the size requirement. In this case, the corresponding component parameters should be altered back in the ADS small signal simulation design platform, and then the above optimization process is followed again. Within one or several iterations, the design goal can be met. This proposed optimization method and platform are used to facilitate designers to efficiently design the class-F output matching network of PA that is close to the actual size with good performance.

### 3.4. Analog Predistortion Technology 

In PA design, when the PA operates in a saturated state, its output signal will generate nonlinear distortion, manifested as nonlinear amplitude distortion and nonlinear phase distortion. Specifically, nonlinear amplitude distortion causes many additional frequency components in the amplifier output spectrum, i.e., spectrum regeneration. These additional frequency components not only cause intermodulation distortion, interception distortion, spurious signals, and adjacent channel interference, reducing the overall performance of the communication system, but also compress the amplifier gain, thereby worsening the efficiency of the PA. On the other hand, nonlinear phase distortion leads to phase lag in the system, which increases the bit error rate at the receiver. In view of the above issues, pre-distortion technology is often used to improve PA linearity. A low-cost and easy-to-implement analog pre-distortion design approach is adopted in this paper that can reduce the cost, volume, computational complexity, and system complexity of the BDS terminal while improving its linearity. In this paper, the input stage amplification circuit of the PA chip is designed as a pre-distortion circuit with gain expansion and phase advance as the input signal power increases, which can compensate for the gain compression and phase lag characteristics of subsequent amplification stages as the input signal power increases. The working diagram is shown in Figure 10. This method can solve the problem of amplitude and phase distortion of PA output signals, improve the linearity of PA, and expand the dynamic range of PA output power. Furthermore, it has many advantages, such as simple structure, low cost, and small size, and does not occupy terminal computing resources.

The schematic diagram of the input amplification stage is the same as Figure 6. As the input signal power increases, the base electrode potential of HBT1 remains unchanged, and the base–emitter voltage of HBT1 Vbe_HBT1 decreases due to its rectification characteristics, resulting in an increase in the current flowing through R1. In this way, the bias circuit can provide HBT0 with a larger base–emitter bias current Ib, making the output power of the input amplification stage higher than the linear output power, thereby enabling the pre-distortion circuit to have gain expansion and phase advance characteristics.

Based on the analog pre-distortion linearization circuit described above, the 3 W/5 W compatible BDS terminal PA chip designed in this paper effectively improves nonlinear amplitude distortion and nonlinear phase distortion. The gain compression point meets the expectation, and the third-order intermodulation distortion (IMD3) and AM-PM distortion are both at high levels in the industry.

### 3.5. Design of Temperature-Insensitive Adaptive Power Detection Circuit

The power detection circuit is a functional module for PA real-time monitoring by the terminal, which is easily affected by the change in ambient temperature. Considering the miniaturization and multi-function requirements of the BDS terminal, to meet the real-time status monitoring of the PA, this paper adopts a compact temperature-insensitive adaptive power detection circuit to achieve a highly reliable power detection function in a limited space.

The schematic diagram of the temperature-insensitive adaptive power detection circuit is shown in Figure 11, which is located behind the second amplification stage shown in Figure 3. In Figure 11, C1 is a small DC block capacitor (1 pF), which forms a power divider with a large capacitor (23 pF) in a 2–3 stage intermediate matching circuit without affecting impedance matching too much [37]. This design allows a portion of the RF signal to enter the power detection circuit through C1 and be rectified by the diodes on the MMIC Die, as well as filtered by the capacitor C2 on the LGA PCB and loaded onto the load resistor R1. The diode rectifier circuit is supplied with a DC bias by the temperature-insensitive adaptive bias circuit shown in Figure 11, whose structure is mostly consistent with the temperature-insensitive adaptive bias circuit in Section 3.2. The only difference of not having a separate Vref power supply port is to reduce power consumption. When the input power or the ambient temperature increases, the bias voltage and current provided by the temperature-insensitive adaptive bias circuit remain relatively stable. Simulation results show that the temperature-insensitive adaptive power detection circuit designed in this paper can achieve a power detection sensitivity of −25 dBm (input power of the PA chip) and a minimum output DC voltage resolution of 1 mV/dBm with a static power consumption of only 160 μA (@5 W) and 97 μA (@3 W).

### 3.6. Thermodynamic Model and Analysis

The PA chip produces a large amount of heat and is significantly affected by ambient temperature changes. In particular, the 3 W/5 W compatible PA chip proposed in this paper has a maximum saturated output power of 38.2 dBm, while the PAE has a maximum value of 51.7%, which means a large amount of DC energy is dissipated in the form of heat. Therefore, in order to ensure the normal operation of the PA chip at ambient temperature without deteriorating the HBT performance due to excessive temperature, it is necessary to simulate the steady-state thermal distribution of the PA chip at ambient temperature. In this paper, ADS is used to simulate the DC power consumption, RF input/output power, PAE, and other characteristic parameters of the chip at various temperatures so that the thermal dissipation power of each transistor can be calculated. Then, ANSYS Workbench is used to load the thermal dissipation power onto the PA chip model. Based on finite element analysis, the chip is decomposed into discrete approximate field elements to solve the temperature distribution nephogram of each region. The results can then be used as assistance in designing the PA chip with a more reasonable layout, improving its overall temperature distribution, lengthening the product life, and increasing stability.

#### 3.6.1. Establishment of Thermodynamic Model for BDS Terminal PA Chip

Firstly, a thermodynamic model of the 3 W/5 W compatible BDS terminal PA chip is constructed based on LGA packaging, whose stack structure is shown in Figure 12a, including MMIC Die, LGA packaging (include PCB), demo board (FR4), and copper carrier at the bottom. Figure 12b is a three-dimensional view of this thermodynamic model, where HL832NXA material is used as a three-layer substrate of the LGA packaging, with a first and third dielectric layer thickness of 30 μm and second dielectric layer thickness of 60 μm. At the surface of each layer, there are copper transmission lines with a thickness of 18 μm. After constructing the model on ADS, the DC power and thermal dissipation power of each amplification stage are simulated and analyzed using current probe I_ Probe, Node Voltmeter, and Power Probe P_ Probe.

Furthermore, based on the size of the HBT, the thermal power density of transistors at the input stage, intermediate stage, and output stage can be calculated. Taking the ambient temperature of 25 °C as an example, the thermal power density calculations for input, intermediate, and output stage HBT transistors are 5.9 × 10^4^ W/mm^3^, 6.8 × 10^4^ W/mm^3^, and 6.5 × 10^4^ W/mm^3^. Finally, with the help of ANSYS Workbench, the overall thermal analysis of the proposed 3 W/5 W compatible BDS terminal PA chip was ultimately achieved through operations such as creating steady-state thermodynamic simulation, setting non-anisotropic properties of materials, meshing, adding thermodynamic loads, and natural convection heat-dissipation surfaces.

The key factor affecting the temperature distribution of a chip is heat transfer, which includes three basic aspects: conduction (Fourier heat conduction formula), convection (Newton’s convection heat transfer formula), and radiation (Stefan–Boltzmann law). The equations are as follows:(13)Q=kA(Thot−Tcold)/δ
(14)Q=aA(Tw−Tair)
(15)$$Q=5.67e^{-8}\varepsilon A(Th^{4}-Tc^{4})$$
where, *k*, *a*, and *ε* are the thermal conductivity, convective heat transfer coefficient, and emissivity, respectively. For these parameters, *a* = 25 W/(m^2^·K), *k* = 377 W/(m·K) for copper used as PCB metal, *k* = 317 W/(m·K) for Au used as DIE metal, *k* = 429 W/(m·K) for Ag used as conducting resin, *k* = 46 W/(m·K) for GaAs used as DIE Substrate, and *ε* = 0.4 for GaAs used as DIE Substrate. In Equation (13), δ is Stefan Pohl’s Hertz constant, about 5.67 × 10^−8^ W/m^2^·k; *A* is the heated area in the solid normal direction, which was selected in ANSYS Workbench; and *Thot* and *Tcold* are the maximum and minimum temperatures solved by ANSYS Workbench. In Equation (14), *Tw* and *Tair* are body temperature and air temperature, solved by ANSYS Workbench. In Equation (15), *Th* and *Tc* are the absolute temperatures of the radiant surface solved by ANSYS Workbench. In Equations (13) to (15), it can be seen that the larger the temperature difference, surface area, and thermal conductivity are, the more heat is transmitted. However, the 3 W/5 W compatible BDS terminal PA chip proposed in this paper is highly integrated, making it difficult to use forced convection heat dissipation (integrated fans, etc.) methods in a limited space. Therefore, when designing the chip layout, the via holes on MMIC Die and LGA PCB are particularly important, whose location and number directly affect the thermal performance of the PA chip, as the generated heat mainly dissipates due to the conduction of metal through holes and the external radiation between substrate layers.

#### 3.6.2. PA Chip Thermodynamic Simulation Auxiliary Design Platform

The thermodynamic simulation auxiliary design platform proposed in this paper is shown in Figure 12c, which aims to provide designers with the actual heating conditions of the chip working under various temperature conditions so that designers can update the layout of the chip in a timely manner or decide whether to add forced cooling convection devices (fans, cooling water, etc.) and other measures to ensure the normal operation of the PA chip and the success rate of chip design and production.

Due to the above purposes, a fully parameterized thermodynamic simulation platform on SCDM (Space Claim Direct Modeler) based on Python is designed in this paper, whose built-in ACT (Application Customization Toolkit) auxiliary programming script is used to achieve parametric modeling of the chip. Furthermore, ANSYS Workbench is used to conduct a CAE (Computer Aided Engineering) thermodynamic simulation through an application programming interface. Then, a temperature nephogram can be derived, facilitating designers to observe the temperature field distribution and modify the packaging structure or layout of transistors, via holes, etc., in a timely manner. Finally, the optimal packaging structure and PCB layout are obtained through genetic algorithm cyclic optimization.

### 3.7. Compatibility Optimization Design Platform of PA Chip

In order to improve the design efficiency of the 3 W/5 W compatible PA chip, an ADS-based PA chip compatibility optimization design platform is proposed in this paper, whose architecture is shown in Figure 13. 

The multiple chips that need to be compatible (the same PA chip at 3.5 V and 5 V supply voltages in this paper) are placed on the same optimization platform (LSSP, Harmonic Balance, and DC simulation controllers in this paper), and the same optimization variables are used, such as capacitance, inductance, resistance, and microstrip line size. The optimization goals are set according to the required compatibility and need to be met simultaneously, such as 3 W PA output of 35 dBm and 5 W PA output of 37.5 dBm. The conventional design approach is to separately simulate PA chips at 3.5 V and 5 V supply voltages, but it is easy to meet the requirements of one PA chip and difficult to meet the other at the same time, requiring multiple iterations to achieve compatible design goals, resulting in low design efficiency. The compatibility optimization platform proposed in this paper effectively improves this situation, and based on this platform, the design cycle of the 3 W/5 W compatible PA chip has been effectively shortened.

## 4. Simulation Results and Analysis

Using the above design, modeling, simulation, and optimization methods on multiple simulation platforms, this paper has designed a 3 W/5 W compatible PA chip for BDS-3 terminals in InGaP/GaAs HBT technology, whose simulated performance parameters are shown in Table 1, and the DC and RF performance parameters and thermal performance simulation results in Table 1 are described in detail below.

### 4.1. DC Characteristics

Based on the ADS simulation platform, the quiescent current of the 3 W/5 W compatible BDS terminal PA chip designed in this paper is 280 mA (@5 V power supply, 5 W output) and 262 mA (@3.5 V power supply, 3 W output), respectively, which is at the leading level in the industry and is very suitable for integration in wireless communication terminals with limited battery capacity such as mobile phones and satellite mobile phones.

### 4.2. RF Characteristics

Based on the ADS platform, the large signal and power characteristics of the 3 W/5 W compatible BDS terminal PA chip are simulated. When the power supply voltages are 5 V and 3.5 V, the chip exhibits absolute stability in the 1.6–1.65 GHz frequency band and input power dynamic range. The input voltage standing wave ratio is less than 1.6:1, and the output voltage standing wave ratio is less than 1.9:1, indicating its well-designed input and output matching circuits. The gain flatness is less than ±0.35 dB. When the power supply voltage is 5 V, the power characteristic simulation results are shown in Figure 14a. The linear gain of the PA at 1.625 GHz (typical value) reaches 39.4 dB; P1dB reaches 37.5 dBm; Psat reaches 38.2 dBm; PAE reaches 51.7%; the second, third, and fifth harmonic suppression ratios are lower than −62 dBc; the AM-PM distortion is less than |2.5°|; and IMD3 is less than −27 dBc. When the power supply voltage is 3.5 V, the simulation results of its power characteristics are shown in Figure 14b. The linear gain of the power amplifier chip at 1.625 GHz reaches 38.7 dB; P1dB reaches 35.1 dBm; Psat reaches 36.2 dBm; PAE reaches 48.2%; the second, third, and fifth harmonic suppression ratios are lower than −65 dBc; the AM-PM distortion is less than |2°|; and IMD3 is less than −25 dBc.

In summary, when the power supply voltage is 5 V and 3.5 V, respectively, the simulation results of the 5 W/3 W compatible BDS terminal PA chip meet all the RF characteristics of the expected 5 W and 3 W outputs, with high output power, high gain, high efficiency, and high linearity.

### 4.3. Thermodynamic Characteristics

The overall layout of the 3 W/5 W compatible PA chip for BDS terminals is optimized using the thermodynamic auxiliary design platform proposed in this paper, where steady-state thermal simulation is performed based on the thermal power density of the chip at a maximum output power of 38.2 dBm (power supply voltage of 5 V). The simulated heat distribution of the PA chip is shown in Figure 15 with an ambient temperature of 25 °C and a natural air convection situation (without any forced-convection heat-dissipation controls).

Under natural air convection conditions, the average temperature of the 3 W/5 W compatible BDS terminal PA chip under saturated operating conditions is about 159 °C, with a maximum temperature of about 198 °C, which is mainly concentrated at the bottom of the transistor and is generally within the safe operating temperature range of the HBT device (according to the characteristics of the HBT, the maximum safe temperature does not exceed 209 °C).

### 4.4. Comparative Analysis

Table 2 lists the performance comparison between the 3 W/5 W compatible BDS terminal PA chip proposed and other similar L band chips. It can be seen that the chip designed in the paper has advantages in terms of PAE, Psat, and gain. Although the size is slightly larger than that of works [24,25], the PAE is much higher.

## 5. Conclusions

Based on InGaP/GaAs HBT technology, using temperature-insensitive adaptive bias technology, class-F harmonic suppression technology, analog pre-distortion technology, temperature-insensitive adaptive power detection technology, and LGA packaging technology, a single MMIC Die integrated 3 W/5 W compatible PA chip for BDS-3 terminals operating in the 1.6–1.65 GHz frequency band using three auxiliary simulation optimization platforms is proposed in this paper. The simulation results show that when the power supply voltage is 5 V, the static operating current of the PA chip is only 280 mA; the linear gain can reach 39.4 dB; P1dB is greater than 37.5 dBm; Psat is 38.2 dBm; the PAE reaches 51.7%; and the second, third, and fifth harmonic suppression ratios are less than −62 dBc. At a power supply voltage of 3.5 V, the static operating current of the PA chip is as low as 262 mA; the P1dB reaches 35.1 dBm; the Psat reaches 36.2 dBm; the PAE reaches 48.2%; and the second, third, and fifth harmonic suppression ratios are less than −65 dBc. It can be seen that the PA chip has high gain, high efficiency, and high linearity characteristics under both output powers, and its various functions and performance are at the leading level in the industry. The chip size is only 6 × 4 × 1 mm^3^, with good thermodynamic characteristics. There are also ESD electrostatic protection circuits and temperature-insensitive adaptive power detection circuits, which can meet the short message function and different application scenarios requirements of battery-powered BDS terminals. Furthermore, this PA chip design provides a reference for similar designs of power amplifier chips that take into account high power, high gain, high efficiency, high linearity, high integration, and thermal stability.

## Figures and Tables

**Figure 1 micromachines-14-01995-f001:**
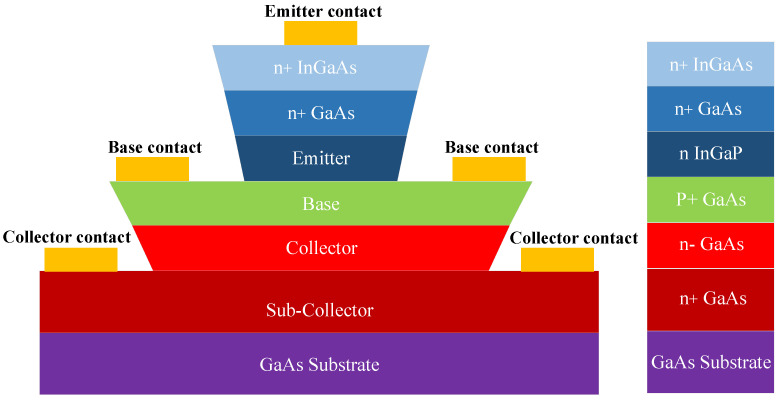
Stacked structure of InGaP/GaAs HBT.

**Figure 2 micromachines-14-01995-f002:**
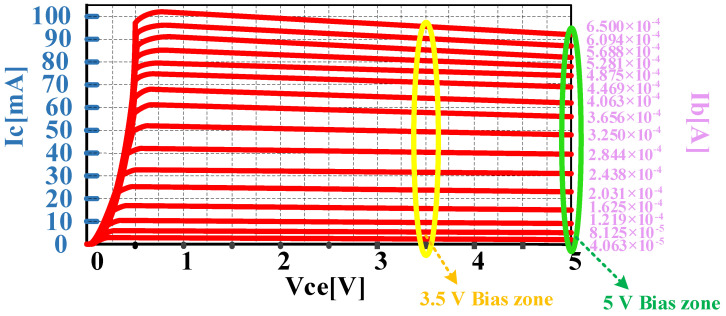
V-I curve of HBT.

**Figure 3 micromachines-14-01995-f003:**
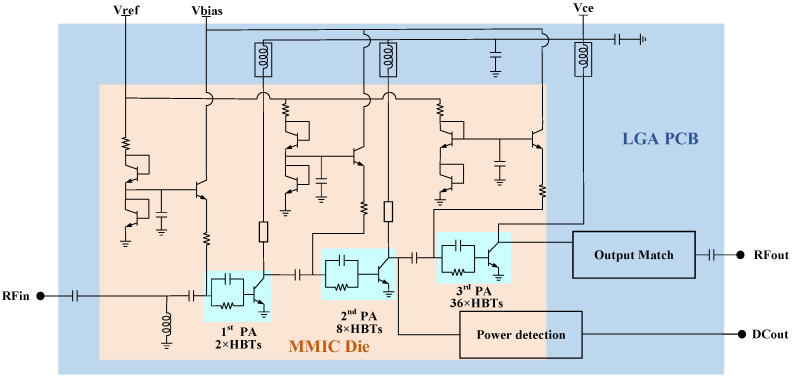
Circuit architecture of 3 W/5 W compatible BDS terminal PA chip.

**Figure 4 micromachines-14-01995-f004:**
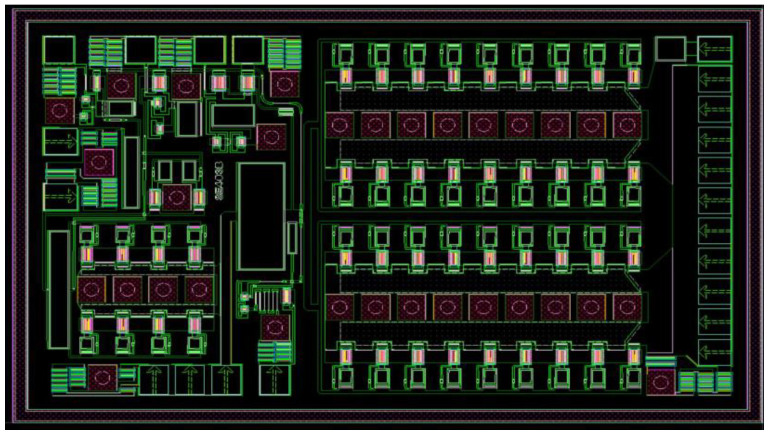
MMIC Die layout of the 3 W/5 W compatible PA chip.

**Figure 5 micromachines-14-01995-f005:**
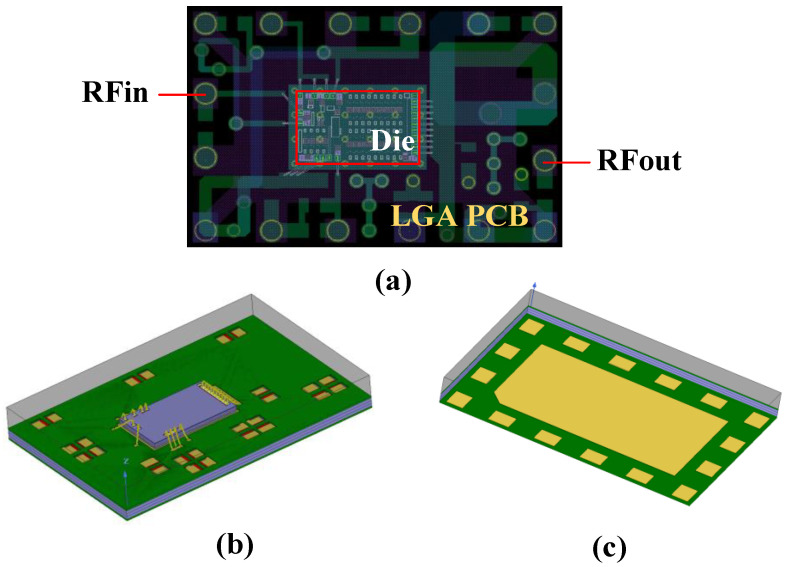
(**a**) The 3 W/5 W compatible BDS terminal PA chip LGA PCB. (**b**) The 3 W/5 W compatible BDS terminal PA chip LGA packaging 3D top view. (**c**) The 3 W/5 W compatible BDS terminal PA chip LGA packaging 3D bottom view.

**Figure 6 micromachines-14-01995-f006:**
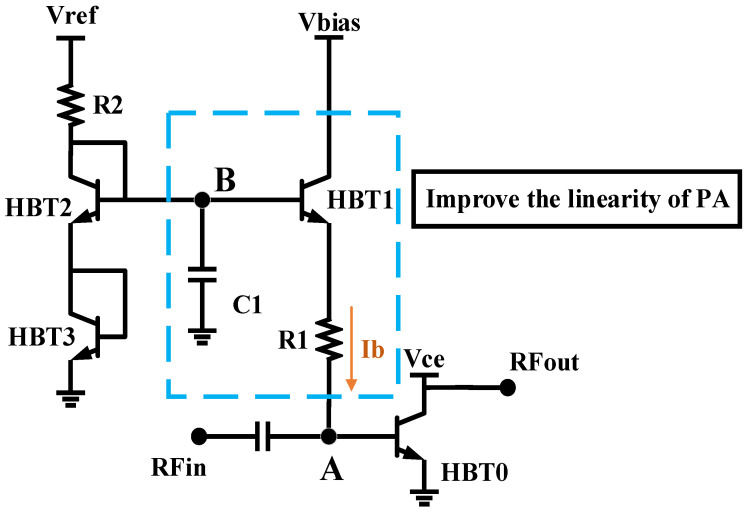
Temperature-insensitive adaptive bias circuit.

**Figure 7 micromachines-14-01995-f007:**
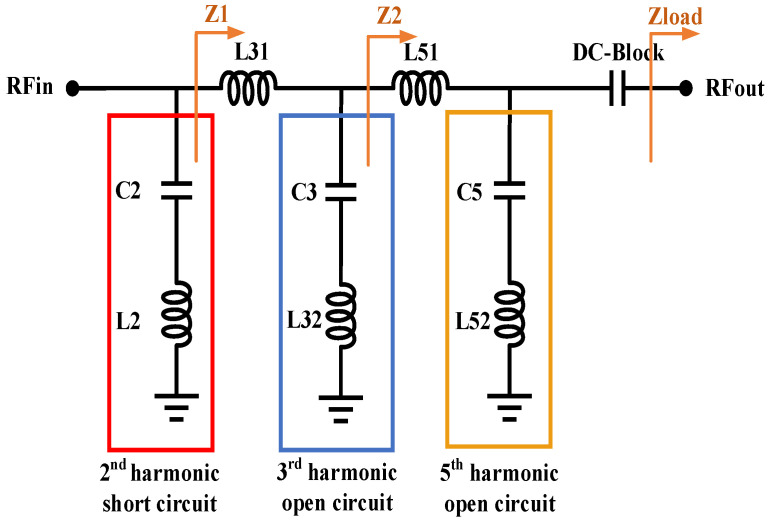
Class-F output matching network with harmonic suppression characteristics.

**Figure 8 micromachines-14-01995-f008:**
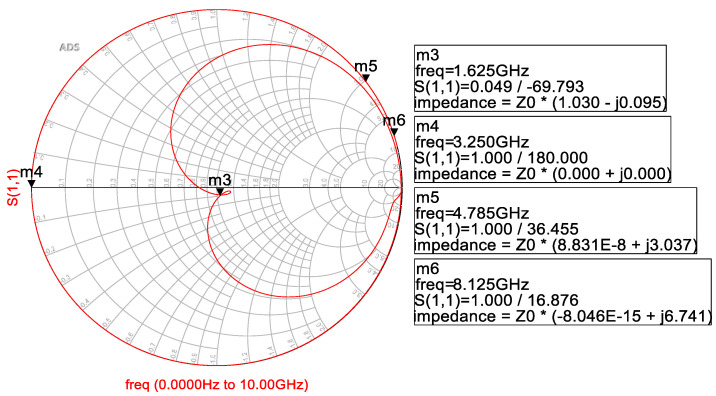
The impedances of fundamental wave and the second, third, and fifth harmonics of the class-F output matching circuit.

**Figure 9 micromachines-14-01995-f009:**
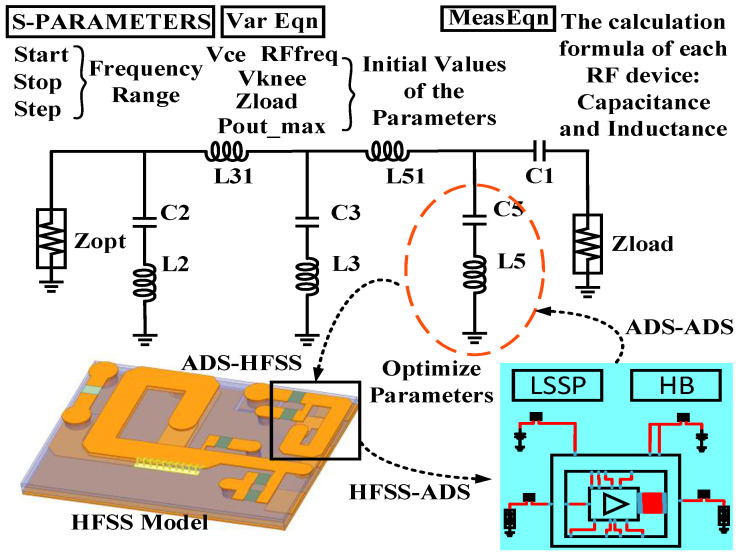
High-efficiency simulation design platform for class-F output matching network based on joint implementation of ADS + HFSS.

**Figure 10 micromachines-14-01995-f010:**
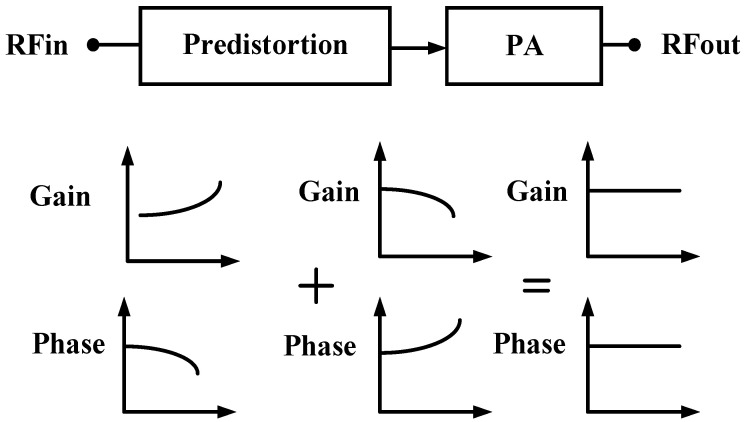
Analog predistortion architecture.

**Figure 11 micromachines-14-01995-f011:**
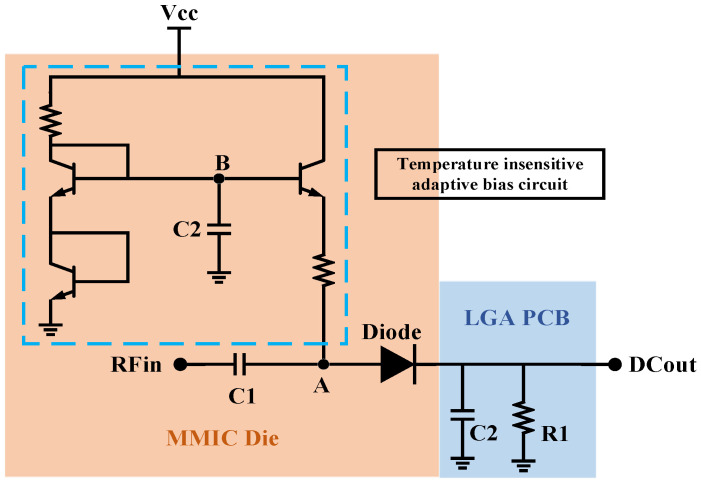
Temperature-insensitive adaptive power detection circuit.

**Figure 12 micromachines-14-01995-f012:**
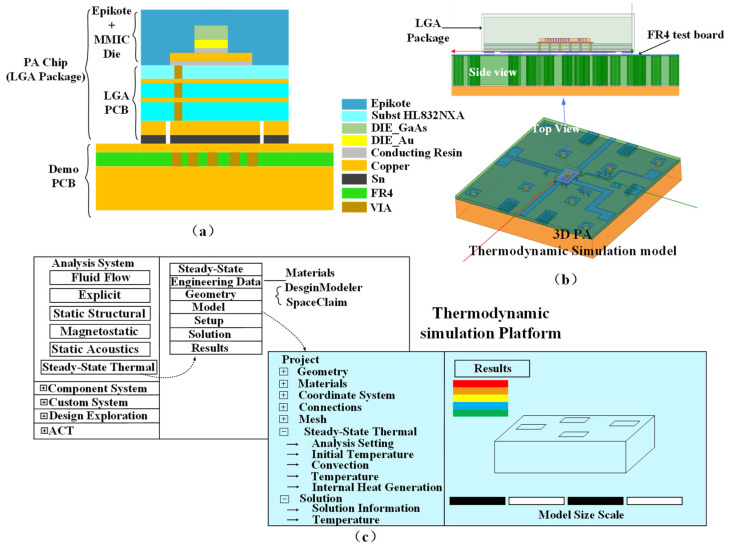
(**a**) Thermodynamic simulation model for 3 W/5 W compatible PA chip on BDS terminal. (**b**) Stack structure of PA chip and Demo PCB. (**c**) Thermal simulation auxiliary design platform.

**Figure 13 micromachines-14-01995-f013:**
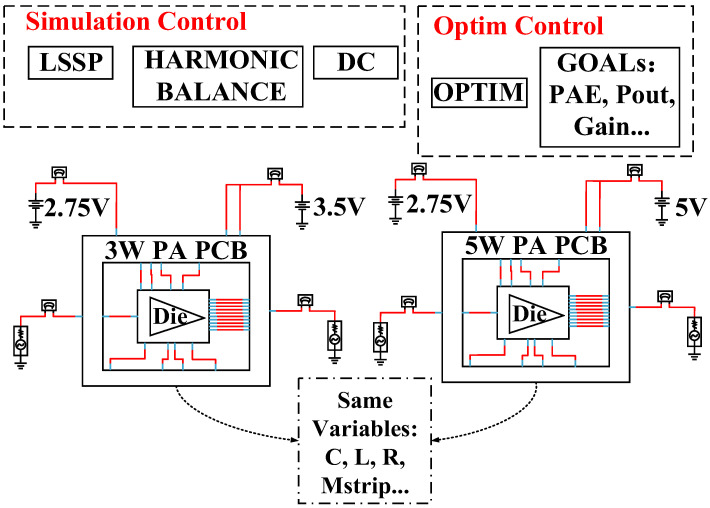
3 W/5 W PA chip compatibility optimization design platform.

**Figure 14 micromachines-14-01995-f014:**
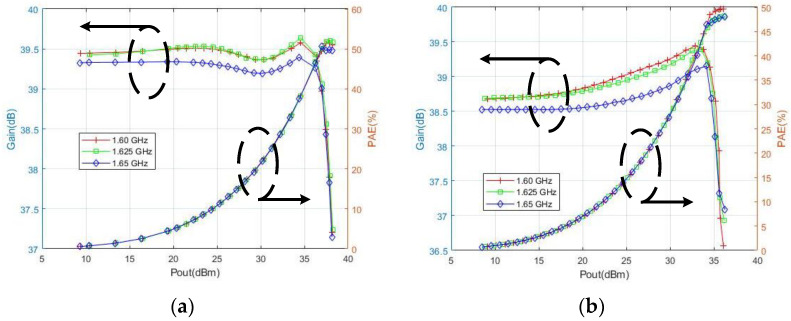
(**a**) Power characteristics of a 3 W/5 W compatible BDS terminal PA chip at 5 V supply voltage and (**b**) power characteristics of a 3 W/5 W compatible BDS terminal PA chip at 3.5 V supply voltage.

**Figure 15 micromachines-14-01995-f015:**
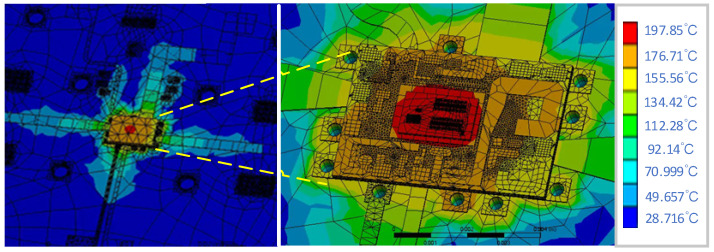
Heat distribution of a 3 W/5 W compatible BDS terminal PA chip under saturated operating state.

**Table 1 micromachines-14-01995-t001:** DC and RF performance parameters of 3 W/5 W compatible BDS-3 terminal PA chip.

Parameters	5 W (5 V Supply Voltage)	3 W (3.5 V Supply Voltage)
Collector–emitter voltage	5 V	3.5 V
I_quiescent_	280 mA	262 mA
Linear Gain	39.4 dB@1.625 GHz	38.7 dB@1.625 GHz
Gain Flatness	≤±0.35 dB	≤±0.35 dB
P1dB	37.5 dBm@1.625 GHz	35.1 dBm@1.625 GHz
P_sat_	38.2 dBm@1.625 GHz	36.2 dBm@1.625 GHz
PAE	51.7%@ Psat @1.625 GHz	48.2%@ Psat @1.625 GHz
IMD3	≤−27 dBc	≤−25 dBc
HSR	−62 dBc@(2f/3f/5f)	−65 dBc@(2f/3f/5f)
AM-PM	|2.5°|@1.6–1.65 GHz	|2°|@1.6–1.65 GHz
VSWRin	1.6:1@50 Ω	1.6:1@50 Ω
VSWRout	1.9:1@50 Ω	1.9:1@50 Ω
Size	6 × 4 × 1 mm^3^	6 × 4 × 1 mm^3^

**Table 2 micromachines-14-01995-t002:** Performance summary of the designed PA chip with other L band designs.

Work	Gain (dB)	P_sat_ (dBm)	PAE	Compatible	Size (mm^2^)
[22]	30	37.8	43%	No	10 × 10
[23]	30	37	45%	No	6 × 6
[24]	38	37.8	36.5%	No	4 × 4
[25]	12	21.3	28.1%	Yes	1.54 × 0.9
	10	26.2	32.2%		
[26]	19.1	17.5	15%	Yes	5 × 5
	31	28	35%		
This work	38.7	36.2	48.2%	Yes	6 × 4
	39.4	38.2	51.7%		

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
