# Peer review of "Design of Power Amplifiers for BDS-3 Terminal Based on InGaP/GaAs HBT MMIC and LGA Technology"

_micromachines, 2023, doi:10.3390/mi14111995_

Round 1
Reviewer 1 Report
Comments and Suggestions for Authors
This manuscript described a PA design based on InGaP/GaAs heterojunction HBT combined with several technologies. Three simulation platform were built to optimize the PA designs. Based on these tools, high gain, high efficiency, and high linearity PA design was achieved. However, the authors did not make it clear the innovation of this work compared to others. The quality of the figures in the article needs to be improved and English writing also needs improvement. The manuscript should be reconsidered after major revision. In addition, there are some issues the authors should also address.
1. The introduction of the background is too long. The authors should refine it.
2. In InGaP/GaAs HBT design part, the authors should clarify more on the reason of the choice of InGaP/GaAs HBT technology.
3. In line 182-183, the authors explained why GaAs is doped with InGaP, corresponding references should be cited here.
4. Fig. 1 is too simple, and it is better to add the materials information to the device structure.
5. In line 203, the authors only provide the emitter area information about the HBT device. Other detailed information like device channel length, active material thickness, contact metal, etc., should be also presented.
6. For equation (13)-(15), not all the parameters were clarified in line 459, including A, T, Thot, etc. The authors should denote them all clearly and the corresponding value used in this paper.
7. Figure 15 has poor resolution, and the temperature value of the legend cannot be seen clearly. The author should provide a high quality one.
Comments on the Quality of English LanguageThe English writing needs to be improved.
Reviewer 2 Report
Comments and Suggestions for Authors
This work presents a power amplifier based on GaP/GaAs technology. The content is interesting and the length is enough for this journal. It is suggested to publish this manuscript after a minor revision.
1. The full expressions should be added to some abbreviations when they appear for the first time in this article, such as MMIC, LGA, CGR and so on.
2. In figure 2,lots of data spectra are presented, it is suggested to point out the assignment of each spectrum.
3. It is difficult to figure out the function of each section of this circuit, please provide some introduction words.
4. It believed that there may be more compatible devices, please add more to Table 2 to improve the convincing.
5. Some wide-bandgap semiconductors literature are suggested to be included, such as Nat Rev Mater 8, 587–603 (2023). https://doi.org/10.1038/s41578-023-00583-9, Appl. Phys. Rev. 8 (2021) 031315, Small Methods 2022, 6, 2101348, Journal of Inorganic Materials, 2023, 38(9): 1055-1061.https://www.jim.org.cn/CN/10.15541/jim20220569
Comments on the Quality of English LanguageMinor revision.
Round 2
Reviewer 1 Report
Comments and Suggestions for Authors
The authors has addressed all the issues. I recommend it publication in its present form.